# Acceptability of a behavioural intervention to mitigate the psychological impacts of COVID-19 restrictions in older people with long-term conditions: a qualitative study

Leanne Shearsmith [1], Peter A Coventry,[2,3] Claire Sloan,[2] Andrew Henry,[2,4] Liz Newbronner [2], Elizabeth Littlewood,[2] Della Bailey,[2] Samantha Gascoyne,[2] Lauren Burke,[2] Eloise Ryde,[2,4] Rebecca Woodhouse,[2] Dean McMillan,[2,5] David Ekers,[2,4] Simon Gilbody,[2] Carolyn Chew-Graham [6]

For numbered affiliations see end of article.

**Correspondence to**
Leanne Shearsmith;
L.Shearsmith@leeds.ac.uk

## ABSTRACT

**Objectives** The COVID-19 pandemic heightened the need to address loneliness, social isolation and associated incidence of depression among older adults. Between June and October 2020, the Behavioural Activation in Social IsoLation (BASIL) pilot study investigated the acceptability and feasibility of a remotely delivered brief psychological intervention (behavioural cctivation) to prevent and reduce loneliness and depression in older people with long-term conditions during the COVID-19 pandemic.

**Design** An embedded qualitative study was conducted. Semi-structured interviews generated data that was analysed inductively using thematic analysis and then deductively using the theoretical framework of acceptability (TFA).

**Setting** NHS and third sector organisations in England.

**Participants** Sixteen older adults and nine support workers participating in the BASIL pilot study.

**Results** Acceptability of the intervention was high across all constructs of the TFA: Older adults and BASIL Support Workers described a positive Affective Attitude towards the intervention linked to altruism, however the activity planning aspect of the intervention was limited due to COVID-19 restrictions. A manageable Burden was involved with delivering and participating in the intervention. For Ethicality, older adults valued social contact and making changes, support workers valued being able to observe those changes. The intervention was understood by older adults and support workers, although less understanding in older adults without low mood (Intervention Coherence). Opportunity Cost was low for support workers and older adults. Behavioural Activation was perceived to be useful in the pandemic and likely to achieve its aims (Perceived Effectiveness), especially if tailored to people with both low mood and long-term conditions. Self-efficacy developed over time and with experience for both support workers and older adults.

**Conclusions** Overall, BASIL pilot study processes and the intervention were acceptable. Use of the TFA provided valuable insights into how the intervention was experienced and how the acceptability of study processes

## STRENGTHS AND LIMITATIONS OF THIS STUDY

⇒ The use of the theoretical framework of acceptability in both informing the topic guide and conducting the analysis, demonstrating a systematic enquiry into acceptability and contributing to the wider field as well as the topic area.

⇒ The length of the interviews facilitated an in-depth exploration of older adults and support workers' experiences.

⇒ Conducting the interviews by telephone while discussing feasibility of telephone delivery may have enabled contextual cues to be discussed that may have been missed in a face-to-face interview set up, however may have led to a self-selecting sample of people who were comfortable with the telephone.

⇒ A limitation is that the short timescale for the study meant that participants had to be interviewed as they completed 3-month outcome measures, rather than using strategic sampling.

and the intervention could be enhanced ahead of the larger definitive trial (BASIL+).

## INTRODUCTION

Loneliness is the subjective psychological expression of social isolation owing to dissatisfaction with the frequency and quality of social contacts.[1] Social isolation is a quantifiable sense of reduced social network size and impoverished social contact.[2] Loneliness is an increasing problem across all age groups, including younger people.[3] However, loneliness is especially problematic in older adults because it is a risk factor for frailty,[4] which in turn is a risk factor for depression.[5] The link between loneliness and subsequent depression can persist for up to 12 years after the loneliness is reported.[6] Social isolation and

loneliness are also established risk factors among older adults for cognitive impairment and mortality.[7 8] In addition to the toll on the individual, loneliness impacts on health and social care services.[9]

In England, the number of people aged≥65 years is projected to grow by 20% over the next decade and by nearly 60% in 25 years.[10] One in four older people in the UK live with a mental health condition, most commonly depression, and the prevalence is higher among older people with multimorbidities.[11] The COVID-19 pandemic and the associated loneliness and lack of social contact have impacted the mental health of older adults.[12] Longitudinal data suggests that loneliness was an important predictor of increase in depressive symptoms in older adults during the pandemic and that the negative mental health impacts of the pandemic persist and may worsen over the long term in the absence of effective interventions.[13] The government's strategy on loneliness sets reducing social isolation and improving well-being among older adults as a public health priority.[14]

The Behavioural Activation in Social IsoLation (BASIL) Study investigated the acceptability and feasibility of a remotely delivered brief psychological intervention (Behavioural Activation) to prevent or reduce depression and loneliness in older people with long-term conditions during isolation (or shielding).[15] The study was adopted by the National Institute for Health and Care Research Urgent Public Health programme in May 2020.[16] Behavioural Activation is an evidence-based psychological treatment. It works on the principle that low mood may be a consequence of physical inactivity and loss of positive reinforcement due to a reduction in valued, pleasurable activities.[17] It is a simpler psychological treatment than cognitive behavioural therapy, but evidence suggests it can be an equally effective therapy for depression, even when delivered by junior mental health workers requiring less intensive and costly training.[18] Emerging evidence suggests that Behavioural Activation has an impact on depression and loneliness in the short term.[19] A bespoke intervention was used in the BASIL Study incorporating Behavioural Activation, a self-help booklet and trained BASIL Support Workers, with the aim of encouraging the older adult to recognise the link between mood and behaviour and reinstate or replace valued activities that generate positive reinforcement and so improve mood.

The BASIL pilot study findings have been reported elsewhere.[15 19] This paper reports the findings from the qualitative study within BASIL, presenting analysis using the theoretical framework of acceptability (TFA).[20]

## METHODS

We conducted a nested qualitative study, using semi-structured interviews to explore the acceptability and feasibility of the intervention from the perspectives of older adult participants and BASIL Support Workers.

## Intervention

Behavioural Activation was adapted with input from our patient and public advisory group. Support workers from a range of clinical and research backgrounds (psychological well-being practitioners, mental health nurses, research assistants, clinical support officers, research nurses and crisis support workers) were trained to deliver the intervention by telephone or video call across eight sessions. The intervention is described in more detail elsewhere.[19]

## Participant selection

In total, 16 study participants were selected via consecutive sampling and interviewed from a pool of 86 study participants from the BASIL pilot study who consented to take part in an interview at initial recruitment (inclusion criteria: aged 65 years or over with two or more physical long-term conditions). Two BASIL pilot study participants did not complete the intervention and one of these participants consented to interview as a non-completer. All nine BASIL Support Workers consented to participate in an interview. Study participants were approached via telephone following completion of the intervention and 1-month post randomisation follow-up measures. Participants were contacted by the researcher (CS) and interviews were arranged at a time to suit the participant. No relationship between participant and researcher was established prior to study commencement. Participants had minimal knowledge about the interviewer, and questions were not provided before the interview. Interviews were conducted by CS, a female research fellow with a PhD in psychology. CS had extensive experience of interviewing and analysis of qualitative data and deaf awareness training. BASIL Support Workers provided their verbal permission to be contacted followed completion of the BASIL intervention training. BASIL Support Workers were provided with study information and invited to contact the interviewing researcher if they were willing to participate in an interview.

## Data collection

One-to-one semi-structured interviews were conducted by telephone. The interviews took place between September and November 2020. Topic guides explored intervention delivery context (eg, impact of the COVID-19 context on older adults), study processes (eg, remote recruitment, mode of intervention delivery) and thoughts around intervention content (eg, study materials). Study topic guides were sensitised by the seven constructs of the TFA[20] (table 1).

Interviews were digitally recorded using an encrypted dictaphone and transcribed verbatim by a professional transcribing company. The 'completer' interviews lasted between 30 and 56 min and BASIL Support Worker interviews lasted between 39 and 60 min. Field notes were not made during or after the interview. Transcripts were not returned to participants.

**Table 1** Overview of TFA domains

| Component | Definition |
| --- | --- |
| Affective Attitude | How an individual feels about the intervention |
| Burden | The perceived amount of effort that is required to participate in the intervention |
| Ethicality | The extent to which the intervention has good fit with the individual's value system |
| Intervention Coherence | The extent to which the participant understands the intervention and how it works |
| Opportunity Costs | The extent to which benefits, profits or values must be given up to engage in the intervention |
| Perceived Effectiveness | The extent to which the intervention is perceived as likely to achieve its purpose |
| Self-efficacy | The participant's confidence that they can perform the behaviour(s) required to participate in the intervention |

The table presents seven-component constructs of the TFA, adapted from Sekhon et al.[20]
TFA, theoretical framework of acceptability.

## Analysis

We initially carried out an iterative thematic analysis (TA),[21] so that we were able to fully explore and familiarise ourselves with the data (reported elsewhere[15]). We then used the theoretical derived TFA,[20] which clearly articulates and defines a range of conceptually distinct constructs related to acceptability of interventions (table 1). These constructs go beyond operational definitions of acceptability such as dropout and uptake rates of interventions which are often more aligned with notions of satisfaction than acceptability. Crucially, the TFA can be used to capture responses about intervention acceptability from both recipients and those involved in delivering interventions. Additionally, the TFA can be used to assess different aspects of acceptability before, during or after intervention delivery, depending on intervention development and evaluation cycle.

Using a theoretically derived framework specifically designed for assessing the acceptability of healthcare interventions was considered appropriate for this study, alongside a TA to provide a multidimensional assessment of the acceptability of the intervention and pilot study processes. We used the TFA after intervention delivery to elicit responses about the anticipated acceptability of the content and delivery of the BASIL intervention with a view to informing refinement of intervention features in advance of definitive evaluation.

Familiarisation of data relevant to the TFA was undertaken by drawing on the meaning of the domains[20] and then indexing data considered relevant to these domains using NVivo V.12. This analysis was carried out using constant comparison[22] methods and regular research meetings were conducted to agree interpretation, relevance and meaning of indexing to the TFA domains. Data were then analysed by CS and AH within the TFA domains, initially by participant group. Constant comparison methods were then applied to compare and contrast data across participant groups (data mapping) by CS. The constant comparison[22] method facilitated the development of categories of phenomena within the TFA domains that allowed for interpretation of participants' perceptions of the acceptability of the BASIL Study. Final data interpretation was agreed through discussion with CS, AH, CC-G and PAC. Participants did not provide feedback on the findings.

## Patient and public involvement and engagement

Stakeholder groups (with older adults, carers, clinicians and third sector practitioners) had been conducted prepandemic to develop the intervention and then online in April 2020 to refine the intervention for remote delivery. The public-facing documents were discussed with our patient advisory group prior to submission for ethics approval. The questions were pilot tested with an individual from the patient advisory group. The findings and analysis from the qualitative study were discussed with the patient participation group.

## RESULTS

### Participants

Semi-structured interviews were conducted with 15 participants who had completed the intervention ('completers') and 1 with a participant who did not complete the intervention ('non-completer'). Data saturation was reached with 15 completer participants,[23] who had between 2 and 8 of a maximum 8 intervention sessions (mean=6.5). Participant demographics for the pilot trial are reported elsewhere[15] along with a summary of the TA (see also online supplemental appendix 1).

Nine interviews were carried out with the BASIL Support Workers who delivered the intervention (characteristics reported elsewhere[15]).

No repeat interviews were carried out.

Themes derived from the data using TA (intervention context, study entry, intervention delivery and content) are presented in online supplemental appendix 1.

### Findings from the TFA analysis

This paper focuses on our findings within the TFA, incorporating both older adult and BASIL Support Worker perspectives. Throughout the paper and online supplemental appendix 1, illustrative data is given with participant identifiers. Older adult intervention 'completer' quotes are labelled with the prefix 'OA', then their identifier (eg, OA09). The intervention 'non-completer' participant is labelled with the prefix 'OA', then 'NC' before their identifier (eg, OA NC07). BASIL Support Worker quotes are labelled with the prefix 'BSW', then their identifier (eg, BSW03). The findings from the TFA analysis are presented under each TFA domain.

### Affective Attitude

Affective Attitude describes 'how an individual feels about the intervention'.[20] Affective Attitude is described

here in terms of the overall retrospective attitude to receiving or delivering the BASIL intervention. Older adults described a positive Affective Attitude towards the BASIL intervention:

> I feel good about taking part. It's definitely been good for me, and I hope, if this gets off the ground, that it's going to be good for other people. It was a benefit for me, it really was a big benefit to me. (OA01)

BASIL Support Workers were positive about being part of the BASIL intervention:

> And I want to make a difference to people, sort of, experience of healthcare and make it more positive and the best, you know, they can have. So, it felt good to be involved in something that was trying to do that. (BSW03)

Some BASIL Support Workers described positive Affective Attitude in terms of adapting the study in response to the COVID-19 pandemic:

> The study was adapted to BASIL to respond to COVID it felt so worthwhile and I felt really motivated to work really, really hard on it. (BSW05)

It appeared, however, that a positive Affective Attitude could be attenuated with study challenges, such as the difficulties with activity planning in the context of COVID-19 restrictions:

> The constant changes in rules and the confusion that people have had over that, has, you know…it has affected it [intervention delivery]. Because one week, someone can be doing okay and the next week, rules have changed and they're feeling anxious, so they're not really sure where they stand anymore. Or, you know, restrictions have been lifted… and they've been able to go back to doing most of what they normally do, and actually, they're loads better. So you really don't know from one week to the next, where someone's going to be at. (BSW07)

Some older adult participants discussed how they had valued the BASIL Support Worker being able to help with problem-solving and information-finding:

> I suppose only that if I did have a problem, there was someone there that I could discuss it with, and someone who could perhaps help me, who knew the right people to contact. (OA13)

### Intervention Coherence

The coherence domain of the TFA is described as 'the extent to which the participant understands the intervention and how the intervention works'.[20] The overall aims of the BASIL Pilot study appeared coherent to both BASIL Support Workers and older adults:

> The aims are to try and help them to find ways of coping with life events such as a pandemic. But not

just that. If you can teach them this behavioural activation cycle thing then they can apply it to any life event or situation that's affecting them and give them a coping strategy. (BSW08)

> I presume, or this is how I looked at it, was that you were trying to get information on how people were reacting within the COVID environment, and that you were hoping to get strategies out of it, that would then help people to cope better with the situation. (OA12)

BASIL Support Workers described the ease with which they could make links between activity and low mood, which underpins the Behavioural Activation part of the BASIL intervention. BSW09 articulates how they think the intervention works, which also informs the 'Perceived Effectiveness' TFA domain:

> So I think it's [BASIL intervention] set up as a structured guided support really, so that they'll get regular contact with one of the support workers who will look to help them to engage with the booklet, and maybe look at how they can make changes to what they're currently doing, in the hope that that's going to impact how they feel physically in terms of their health, but also emotionally. (BSW09)

Older adults appeared to have a good understanding of the relationship between activity, mood and physical health, suggesting that the intervention was understandable to participants and had a reasonable level of face validity:

> I think they link greatly because obviously if you're doing something that keeps your…makes you feel good and keeps your mind away from anxieties then that's obviously going to help you feel better, it's going to help you face each day in a better way. (OA09)

However, it appeared that the coherence of the intervention could be reduced when using Behavioural Activation with participants who were not experiencing low mood:

> Some [older adults] just said because they didn't have a low mood it didn't affect their behaviour. It was very difficult to try and make them see the cycle when they didn't see any relevance for them because they were fine. (BSW08)

BASIL Support Workers reflected that when people did not perceive themselves as having low mood, the intervention made less sense to them and they did not see how Behavioural Activation could work for them.

### Opportunity Cost

Opportunity Cost describes the 'extent to which benefits, profits or values must be given up to engage in an intervention'.[20] Most BASIL Support Workers talked about how their usual working role had been adjusted to accommodate the delivery of the intervention. As such

intervention delivery had little cost to their usual working role:

> P: Yeah, my caseload was adjusted slightly, so I had a couple of client contacts less my caseload for the week to create the time to be able to work on the study.
>
> I: Did that work okay?
>
> P: Yeah, it was absolutely fine, I was really supported by my [work] to be able to do that, they've been brilliant. (BSW04)

Other BASIL Support Workers described how intervention delivery fitted in with their working routine: *I just fitted it in with my diary and that worked fine* (BSW07).

Generally, older adults described a low Opportunity Cost in terms of taking part in the intervention. This was attributed, in part, to having time available: *It wasn't bothering my timetable or anything like that, cause with the lockdown I've been doing very little anyway* (OA11). Other older adults described how the intervention could be planned flexibly around their other activities:

> I did, just work it in and around my life. If there was something that had to be wrote down, then I would sort of spend perhaps an hour thinking about it, and then I'd go on and do something else, and then I'd come back to it, and spend another hour maybe writing down what I wanted to say and that, you know. (OA12)

### Burden

The TFA domain of Burden describes 'the perceived amount of effort that is required to participate in the intervention'.[20] Some BASIL Support Workers described a time Burden associated with delivering the intervention sessions:

> If somebody lives on their own and the social isolation is more of a problem, then it's really hard to keep it to 30 minutes and generally they take an hour, those calls. (BSW06)

One BASIL Support Worker discussed the emotional effort involved in delivering the intervention in the pandemic, due to the social restrictions that everyone faced, and reported that it could be an effort to stay positive about the future:

> It was just everything was just a bit more of an effort but that was the world per se really. It was for everybody and anybody that was working everything was more difficult, yeah. (BSW01)

Some older adults felt the amount of effort required to take part in the BASIL Study was minimal, while others described the mental effort involved in taking part in the intervention. The data extracts below illustrate these contrasting views:

> There was a little bit of effort. I mean obviously from week to week you had these tasks, your diary to fill in and to think about the tasks that you were doing that weren't enjoyable and how I could break it down into smaller chunks rather than do the whole thing in one go…So there were some things to do but it wasn't onerous at all. (OA03)

> No physical effort whatsoever. Mental effort is the problem. And just, I didn't want to let the [person] who was doing the questioning, down, so I used to have to think about what was required, you know, and what [they] was expecting from me, and then try and adapt it into it. So, yes, there was a bit of effort involved, from the point of view that you don't want to do it, but you have to do it, to keep things on an even track, you know? (OA12)

### Self-efficacy

The Self-efficacy domain of the TFA describes 'the participant's confidence that they can perform the behaviour(s) required to participate in the intervention'.[20] Here, Self-efficacy is described in terms of how confident both BASIL Support Workers and older adults felt undertaking the BASIL intervention overall.

Some BASIL Support Workers described how, over time, their confidence to deliver the intervention grew with experience:

> I think your confidence definitely grows with familiarity, so the first few times you're having a session with a participant, you're very keen to make sure you've got everything, all your materials to hand, that you know what you're doing and what's expected of you, making sure that you're following the protocol. As it all starts to become more familiar and you can really enjoy using the booklet, it's really nice to have such nice materials to use with the client and the participant. (BSW04)

Some participants expressed a degree of increased confidence, satisfaction and mastery in having completed the intervention:

> I wouldn't say confident, but I did learn a lot, and I think at the end of it, I did feel more confident. At the end of it all, I thought, well, I've done that and I'm much better off for doing it. (OA01)

Others were, however, more equivocal about engaging with the intervention but still recognised that they had become more certain about coping with the pressures of social isolation:

> I didn't feel unconfident. I felt apprehensive, maybe, you know, it's all this…is this really Big Brother keeping their eye on us, blah blah blah. But I think everybody I've spoken to has been really friendly and helpful and it's helped draw me out and helped me see things better and cope with things better. No, I

don't think I felt unconfident, just more apprehensive. (OA14)

## Perceived Effectiveness

Perceived Effectiveness describes 'the extent to which the intervention is perceived as likely to achieve its purpose'.[20] During the interviews, we asked if participants thought the BASIL Study would achieve its aims, and to gauge how positive they felt about this, we also asked if participants would 'recommend the study to others'.[24]

Older adults reflected that the BASIL Study would achieve its aims:

It does cover quite a lot really. As I say, it seems to be, you know, centred obviously towards mood and how the…this pandemic has affected how people feel and there, sort of, fears and what have you. So yes, I think it's very good. (OA09)

BASIL Support Workers also acknowledged that Behavioural Activation had the potential to support people who were shielding and isolated during the pandemic:

I hadn't thought about it, in terms of delivering for COVID. And I think, yeah, it's a really good approach for this. Obviously, I've never thought about it before, I've not been in a global pandemic before, but yeah, I think it's a useful intervention. (BSW07)

Both older adults and BASIL Support Workers reflected that the effectiveness of the BASIL intervention was likely to be enhanced if it was more specifically targeted at those with low mood and long-term physical health problems:

I think possibly it needs to be more targeted, so anybody who has painful medical condition or who lives alone who is isolated, certainly I think it would benefit them a lot. I think the wide spread that you've currently can be more targeted and more focused and more helpful to more people in that sense. (OA02)

I think, yes, but I do think just some consideration needs to be given to who we're targeting, maybe it's not quite so useful for people on the threshold of depression and feel that they're doing quite well……….
I wonder if perhaps it would be more beneficial just to focus on people who are expressing their own thoughts of feeling low. (BSW04)

Most participants (Support Workers and older adults) said they would recommend the BASIL Study to others:

Yes, definitely. But either I'd like to think that perhaps an older family member who was struggling and there was an opportunity, who was struggling who knew there was an opportunity to participate in the BASIL study, I would say yes. I'd also say yes to my colleagues because I think it really helps. (BSW04)

I think, well, I'd definitely recommend it to people who want to gain some experience with working with older adults, because it's great insight. And then for

the older adults themselves, yeah, particularly those who are struggling with low mood, I can see that it is a really valuable resource and it's helpful to have that kind of contact, even if it just gets them talking about how they're feeling, because that might not be something that they've done previously. (BSW09)

## Ethicality

The Ethicality subtheme describes 'the extent to which the intervention has a good fit with an individual's value system'.[20] Ethicality was explored by enquiring what participants valued most or least about the BASIL intervention. BASIL Support Workers often described how they had valued being able to help older people during the pandemic:

I think most valid was just getting to know and help older adults, particularly at this time where we know mental health, particularly things like depression and things like that, has risen. So that's been [inaudible 53:15], I just like the fact that I was able to help this group of people even if it is only [number] of them I've managed to help. (BSW 06)

Other older adults described feeling positive about the BASIL intervention, primarily because it presented an opportunity to be altruistic:

I: How do you feel about taking part in the BASIL Support Package?

P: I think if it's helping the likes of yourself to understand how people are managing that if I've been able to be a part of that and assist in any way I'm pleased to have done it. (OA07)

One older adult appeared to value the social contact the intervention provided:

I think, it's actually nice to speak to people that are doing this because I think it's…in the present situation, I think anybody who's doing something like you are doing, I think is pretty good, I think it's wonderful and it's great that you're doing it, you know, for whatever reason. (OA09)

Others discussed personal gains made from taking part in the intervention overall:

The thing I valued most is that it has made me more reflective and more grateful and that's the thing that I got out of it that it made me really look at my life and re-think how I do things. (OA02)

## DISCUSSION

The BASIL Pilot study was designed to test the feasibility and acceptability of recruiting and remotely delivering a Behavioural Activation intervention to older adults with two or more long-term conditions during the COVID-19 pandemic. The intervention was set within a collaborative care framework and was designed to prevent and reduce

depression and loneliness in socially isolated older adults. The initial thematic analysis (TA) (online supplemental appendix 1) suggested that the recruitment procedures to the BASIL pilot trial and the intervention (including the self-help booklet) were acceptable. Findings from the TA[21] informed the protocol for the definitive BASIL+ randomised controlled trial.[25] Remote delivery of the intervention was acceptable and the self-help booklet was engaging and relevant, but less so for those without low mood. Activity planning during the intervention was difficult while COVID-19 restrictions were in place. While altruism facilitated study participation, the BASIL trial needs to be targeted to those with depressive symptoms. TA highlighted detail surrounding the intervention context, variability in how the restrictions had impacted participants' mental health and perceived lack of access to primary care which could influence the way participants felt about the intervention but were not picked up by the TFA.

The TFA[20] added to the TA analysis and provided a comprehensive framework to inform the retrospective and prospective acceptability of the intervention. All seven of the TFA component constructs were populated with data from the interview transcripts and provided insights that were not identified by the initial TA. However, the Ethicality component was more difficult to code and sometimes overlapped with the Affective Attitude component. Confidence in the intervention (Self-Efficacy) grew over time for both older adults and BASIL Support Workers. For older adults, previous experience of similar interventions promoted Self-efficacy, and some older adults reported greater Self-efficacy as they experience positive effects from taking part. For BASIL Support Workers, Self-efficacy grew with experience of delivering the intervention, although was reduced by having a long gap between training and intervention delivery. Both older adults and Support Workers valued the opportunity to demonstrate altruism in the context of the pandemic, but older adults also valued the social contact and assistance with problem solving and seeking healthcare provision. The intervention had low Opportunity Cost for both Support Workers and older adults, who found delivering and taking part in the intervention fitted in around other commitments.

### Theoretical constraints of TFA

Burden and Opportunity Cost were the least populated domains—it could be that low perceived Burden and low Opportunity Cost means participants elaborate less on these topics or it could be that the volume and richness of the data coded to other domains indicates that those elements appealed more to participants compared with the Burden or Opportunity Cost of the BASIL intervention. Ethicality was a difficult component to evaluate and was closely related to Affective Attitude. The questions 'Is there anything you valued about the BASIL Study intervention?' and 'Would you recommend the BASIL Study to others?' were used to target Ethicality, but it

was sometimes difficult to separate participants talking about what they valued versus talking about their value system; often participants described liking the intervention for what it can offer and the benefits they gained, and it was left to the research team to interpret whether this was in line with their value system or an expression of how they felt. Perhaps, Ethicality is easier to analyse in interviews with healthcare professionals with a clear set of professional values, compared with interviews with participants where their individual value system and core personal values are more implicit. Future iterations of the TFA should consider elaborating on the construct meanings and perhaps provide example questions that researchers can adopt in topic guides. The TFA approach has added value, justification for concluding that the intervention is acceptable, provided insights into how to refine the recruitment processes for a definitive main BASIL trial (BASIL+ ISRCTN63034289) and is comprehensive enough to evaluate acceptability alone if time and resources are limited, however, carrying out an inductive TA first elucidated important contextual information that would have otherwise been missed, therefore we would recommend doing both.

### Strengths and limitations

This study was conducted as an urgent public health study during the pandemic, the intervention and data collection were adapted in line with government restrictions. The use of TA followed by TFA informed the topic guide and strengthened our findings, demonstrating a systematic enquiry into acceptability and contributing to the wider field as well as the topic area. A limitation is that conducting telephone interviews may have led to a self-selecting sample of people who were comfortable with the telephone. A low intervention dropout rate of 2% in the BASIL pilot study[15] meant that there were low numbers of non-completers available to include in the qualitative study. A higher dropout rate would have enabled a larger sample of participants who did not complete the intervention. Telephone delivery of the intervention was acceptable, although some older adults suggested that if it were not for the pandemic, they would have preferred face-to-face delivery. This finding fits with previous work in the acceptability of telephone-based therapy for depression. Despite user ambivalence, there is no clear evidence that the use of the telephone negatively affects the interactional elements of therapy.[26] Furthermore, telephone-delivered case management has proven efficacy in supporting the implementation of multidisciplinary approaches to managing depression, such as collaborative care.[27] Our findings offer further support for the utility of using the telephone to deliver low-intensity psychological interventions, even where there might be an expressed preference for face-to-face approaches.

To conclude, this nested qualitative study explored the perspectives and experiences of those involved in delivering and participating in the BASIL pilot study.[15]

Analysis using the TFA[20] provides a novel analytic lens to understand key aspects of acceptability, but an initial TA is valuable. To our knowledge, this is the first process evaluation of a trial and intervention designed to mitigate the psychological impact of COVID-19, addressing the research priority of evaluating brief psychosocial interventions to prevent depression and loneliness in vulnerable populations during the COVID-19 pandemic.[28] This contributes to the evidence around psychosocial and behavioural interventions for older adults addressing the mental health impact of COVID-19 and beyond.

**Author affiliations**
[1]Leeds Institute of Health Sciences, University of Leeds, Leeds, UK
[2]Department of Health Sciences, University of York, York, UK
[3]York Environmental Sustainability Institute, University of York, York, UK
[4]Research and Development, Tees Esk and Wear Valleys NHS Foundation Trust, Flatts Lane Centre, Middlesbrough, UK
[5]Mental Health and Addiction Research Group, Hull York Medical School, Hull, UK
[6]School of Medicine, Keele University, Keele, UK

**Acknowledgements** The authors would like to thank the participants for taking part in the trial and our Patient and Public Advisory Group members for their insightful contributions and collaboration.

**Contributors** LS and CS contributed to the writing of the manuscript. CS contributed to data acquisition. CS and AH contributed to data curation. CS, CC-G and PAC contributed to data analysis. LS, CC-G, PAC, LN and CS contributed to the creation and presentation of the data. EL, SGi and DE contributed to data validation. LS, CC-G, PAC, LN, RW and CS contributed to the editing and critical review of the manuscript. LS, CC-G, PAC, EN, RW, CS, AH, SGi, DE, DM, ER, LB, SGa and DB contributed to the planning, conduct and review preparation. CC-G, PAC, CS, EL and SGa contributed to the conceptualisation of the overarching research aims. SGi, DE, CC-G, EL, PAC and DM contributed to funding acquisition. CC-G, PAC, CS, EL, SGa, LB, ER, AH, LS, DB and DM contributed to methodological design and development. CS, EL and CC-G contributed to project administration. CC-G, DE, SGi, EL and CS contributed to study oversight and leadership. LS is acting as guarantor for the work.

**Funding** This work was funded by the National Institute for Health and Care Research (NIHR) under its Programme Grants for Applied Research (RP-PG-0217-20006). The funder had no role in study design, data collection and analysis, decision to publish or preparation of the manuscript. The views expressed are those of the author(s) and not necessarily those of the NIHR or the Department of Health and Social Care.

**Competing interests** DE and CC-G are current committee members for the NICE Depression Guideline (update) Development Group and SGilbody was a member between 2015 and 2018. SGilbody, PAC and DM are supported by the National Institute for Health and Care Research Yorkshire and Humberside Applied Research Collaboration (ARC), and DE is supported by the North East and North Cumbria ARCs.

**Patient and public involvement** Patients and/or the public were involved in the design, or conduct, or reporting, or dissemination plans of this research. Refer to the Methods section for further details.

**Patient consent for publication** Not applicable.

**Ethics approval** This study involves human participants and was approved by Yorkshire and The Humber - Leeds West Research Ethics Committee ref: 18/YH/0380 (approved as substantial amendment 02 under existing NIHR IRAS249030 research programme). The protocol for the BASIL pilot study was preregistered (ISRCTN94091479) on 9 June 2020. Participants gave informed consent to participate in the study before taking part.

**Provenance and peer review** Not commissioned; externally peer reviewed.

**Data availability statement** All data relevant to the study are included in the article or uploaded as supplementary information.

**ORCID iDs**
Leanne Shearsmith http://orcid.org/0000-0002-6575-0529
Liz Newbronner http://orcid.org/0000-0003-2366-9981
Carolyn Chew-Graham http://orcid.org/0000-0002-9722-9981

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
