## [Reviewer comments · BMJ Open]

ARTICLE DETAILS

TITLE (PROVISIONAL)	Acceptability of a behavioural intervention to mitigate the psychological impacts of COVID-19 restrictions in older people with long-term conditions: a qualitative study
AUTHORS	Shearsmith, Leanne; Coventry, Peter; Sloan, Claire; Henry, Andrew; Newbronner, Liz; Littlewood, Elizabeth; Bailey, Della; Gascoyne, Samantha; Burke, Lauren; Ryde, Eloise; Woodhouse, Rebecca; McMillan, Dean; Ekers, David; Gilbody, Simon; Chew-Graham, Carolyn

VERSION 1 – REVIEW

REVIEWER	Jackson, Alun Australian Centre for Heart Health,
REVIEW RETURNED	18-Jul-2022

GENERAL COMMENTS	This paper achieves its aim of presenting data from an embedded qualitative study within the BASIL pilot study, to inform full protocol development by assessing acceptability using thematic analysis of semi-structured interviews and TFA analysis. A minor point: Although the thematic data are presented in Appendix 1, the themes could be named at line 52 of page 8.
---

REVIEWER	Smith, Kimberley University of Surrey, FHMS
REVIEW RETURNED	05-Sep-2022

GENERAL COMMENTS	Overall, this is an interesting paper where participants (and support workers) in a behavioural activation study speak about their experiences of taking part in the study. The authors have already published a thematic analysis of these results, but this time have analysed the data using the theoretical framework of acceptability. 1.) There are a lot of acronyms in the abstract and throughout the article which do make it difficult to navigate. I would suggest keeping acronyms to a minimum in order to improve readability (maybe 2-3 key acronyms). 2.) From the abstract I had thought you would include both your thematic analysis and TFA analysis, whereas upon reading your paper it becomes obvious that you just present the TFA analysis (as the other qualitative study has been published elsewhere). Within the abstract could you make it clearer that this is an analysis based only on the TFA alongside a short definition of what this is (I found it hard to reconcile your results to the TFA
--

	without knowing what this was). I found the results to read quite simplistically and descriptively and think that they could be stronger if you just really clearly map onto TFA domains. 3.) In the introduction you state that ageing is a risk factor for loneliness and social isolation. Within the loneliness literature there is an acknowledgement that loneliness can affect anyone across the lifecourse, and that often we observe higher rates of loneliness in younger people. Stating that loneliness is linked to older age perpetuates an unhelpful societal and ageist narrative that loneliness is a problem of older age. I know this is a very long comment for a small part of your paper, but I think it would be helpful if you were to frame this a little differently. 4.) The introduction is very short and doesn't really provide a compelling context for your study. From reading this someone could assume your intervention was focused mostly on loneliness and social isolation, whereas you actually focus on loneliness and depression. Could you provide a more detailed overview of both loneliness and depression in the context of older adults living with LTC's during COVID-19, what behavioural activation is, and why behavioural activation could help improve loneliness and depression? It would also be good to have more of an overview of the TFA and how that helps us to assess the acceptability of intervention studies within the introduction too (and has this been used previously in other studies for the same reason?) 5.) Could you also provide more information within the methods on how behavioural activation was adapted in this study, who delivered the programme etc. 6.) Within your methods could you provide more information about how the initial 86 people were recruited, how many people agreed etc (it would be good to know for transparency how many people were initially approached and agreed to take part). 7.) Within the results could you reflect a little more on the strengths and limitations of your work? For instance, the vast majority of people you interviewed had completed the study - do you think results could have looked different had you interviewed more people who did not complete the intervention?
--	--

VERSION 1 – AUTHOR RESPONSE

Reviewer comment	Response
Reviewer 1	
This paper achieves its aim of presenting data from an embedded qualitative study within the BASIL pilot study, to inform full protocol development by assessing acceptability using thematic analysis of semi-structured interviews and TFA analysis.	We thank the reviewer for this positive feedback.

Although the thematic data are presented in Appendix 1, the themes could be named at line 52 of page 8.	Thank you for this comment. Line 52 of page 8 is mid-paragraph in the Affective attitude – TFA results. We have incorporated the reviewer’s suggestion by including the theme names in the results section before the TFA analysis.
Reviewer 2	
Overall, this is an interesting paper where participants (and support workers) in a behavioural activation study speak about their experiences of taking part in the study. The authors have already published a thematic analysis of these results, but this time have analysed the data using the theoretical framework of acceptability.	We thank the reviewer for the positive feedback and for taking the time to review our paper. We have done our best to address your comments and suggestions as detailed below.
1.) There are a lot of acronyms in the abstract and throughout the article which do make it difficult to navigate. I would suggest keeping acronyms to a minimum in order to improve readability (maybe 2-3 key acronyms).	We have reduced acronyms (BA, BSWs and LTCs have been removed) leaving 2 key acronyms (TFA and BASIL).
2.) From the abstract I had thought you would include both your thematic analysis and TFA analysis, whereas upon reading your paper it becomes obvious that you just present the TFA analysis (as the other qualitative study has been published elsewhere). Within the abstract could you make it clearer that this is an analysis based only on the TFA alongside a short definition of what this is (I found it hard to reconcile your results to the TFA without knowing what this was). I found the results to read quite simplistically and descriptively and think that they could be stronger if you just really clearly map onto TFA domains.	We appreciate the reviewer’s request for more clarity that this is an analysis based on the Theoretical Framework of Acceptability (TFA) rather than thematic analysis. We agree that the focus of the paper is on TFA, however, we have included some thematic analysis findings to demonstrate that we conducted thematic analysis first before TFA. The thematic analysis findings are described in the first paragraph of the discussion, and included in the paper in Appendix 1 (there is a sentence in the results section signposting the reader to this). Therefore we have decided to keep thematic analysis in the design section of the abstract as this reflects what is reported in the manuscript. There isn’t capacity within the 300 word limit of the abstract to add a definition, however we have taken the reviewer’s suggestion of providing more clarity on what the TFA is. In addition to the outline of TFA domains already provided in table 1, we have expanded our description of the TFA in the Analysis section of the Methods and clarified how and why we’ve applied the TFA as a framework. While we appreciate the reviewer’s suggestion to clearly map onto TFA domains, we believe that presenting our results using the 7 domains of the TFA is sufficiently clear. We have used subheadings for each TFA domain with quotes and analysis mapped onto each. In terms of simplicity and descriptiveness, this is

	an applied piece of qualitative research and we're targeting this paper to a readership that wants to understand the acceptability of the intervention.
3.) In the introduction you state that ageing is a risk factor for loneliness and social isolation. Within the loneliness literature there is an acknowledgement that loneliness can affect anyone across the life course, and that often we observe higher rates of loneliness in younger people. Stating that loneliness is linked to older age perpetuates an unhelpful societal and ageist narrative that loneliness is a problem of older age. I know this is a very long comment for a small part of your paper, but I think it would be helpful if you were to frame this a little differently.	This is an important point, and we agree that loneliness can affect anyone across the life course. In order to frame this differently, we have removed "aging is a risk factor for social isolation" and included a reference to show that loneliness is a problem across all age groups, including younger people.
4.) The introduction is very short and doesn't really provide a compelling context for your study. From reading this someone could assume your intervention was focused mostly on loneliness and social isolation, whereas you actually focus on loneliness and depression. Could you provide a more detailed overview of both loneliness and depression in the context of older adults living with LTC's during COVID-19, what behavioural activation is, and why behavioural activation could help improve loneliness and depression? It would also be good to have more of an overview of the TFA and how that helps us to assess the acceptability of intervention studies within the introduction too (and has this been used previously in other studies for the same reason?)	We thank the reviewer for this comment. We are constrained by the word limit here, as this article was already 500 words over the word limit at original submission. Expanding the introduction will lead the manuscript further over the word limit. Nonetheless, to respond to the reviewer's comments, we have added a sentence in the introduction to show that behavioural activation is an evidence-based psychological treatment. We have also included two sentences with references to show the rationale for using behavioural activation. We have included a sentence referencing longitudinal data on loneliness and depression in the context of older adults during COVID-19. In terms of the TFA, we have responded to this in point 2 above. We hope this is acceptable to the reviewer.
5.) Could you also provide more information within the methods on how behavioural activation was adapted in this study, who delivered the programme etc.	We have taken the reviewer's suggestion and added an Intervention subheading in the methods, describing how behavioural activation was adapted and who delivered it. Since submitting this manuscript the 12-month BASIL pilot study results have been published which includes more detailed description of the intervention, and we have included a reference for this. We have tried to avoid duplicating information that is available elsewhere due to word count constraints.
6.) Within your methods could you provide more information about how the initial 86 people were recruited, how many people agreed etc (it would be good to know for	References 12 and 26 are the BASIL study protocol and BASIL pilot results which includes this detail. We would like to highlight to the

transparency how many people were initially approached and agreed to take part).	editor that duplicating these details here will further increase the word count.
7.) Within the results could you reflect a little more on the strengths and limitations of your work? For instance, the vast majority of people you interviewed had completed the study - do you think results could have looked different had you interviewed more people who did not complete the intervention?	We thank the reviewer for prompting us to reflect further on the strengths and limitations of our work. We have added a paragraph in the discussion. We had low numbers of non-completers because there was a low drop-out rate in the BASIL pilot study (2%, 2 out of 47 in the intervention arm). We have added a sentence in the methods section to show that 2 participants did not complete the BASIL pilot study intervention and of these, one consented to interview.